# Repeated information of benefits reduces COVID-19 vaccination hesitancy: Experimental evidence from Germany

**Maximilian Nicolaus Burger**⬡*, **Matthias Mayer**⬡, **Ivo Steimanis**⬡

Department of Economics, Philipps University Marburg, Marburg, Germany

⬡ These authors contributed equally to this work.
* maximilian.burger@wiwi.uni-marburg.de

## Abstract

### Background

Many countries, such as Germany, struggle to vaccinate enough people against COVID-19 despite the availability of safe and efficient vaccines. With new variants emerging and the need for booster vaccinations, overcoming vaccination hesitancy gains importance. The research to date has revealed some promising, albeit contentious, interventions to increase vaccination intention. However, these have yet to be tested for their effectiveness in increasing vaccination rates.

### Methods & results

We conducted a preregistered survey experiment with N = 1,324 participants in Germany in May/June 2021. This was followed by a series of emails reminding participants to get vaccinated in August and concluded with a follow-up survey in September. We experimentally assess whether debunking vaccination myths, highlighting the benefits of being vaccinated, or sending vaccination reminders decreases hesitancy. In the survey experiment, we find no increase in the intention to vaccinate regardless of the information provided. However, communicating vaccination benefits over several weeks reduced the likelihood of not being vaccinated by 9 percentage points, which translates into a 27% reduction compared to the control group. Debunking vaccination myths and reminders alone also decreased the likelihood, yet not significantly.

### Discussion

Our findings suggest that if soft governmental interventions such as information campaigns are employed, highlighting benefits should be given preference over debunking vaccination myths. Furthermore, it seems that repeated messages affect vaccination action while one-time messages might be insufficient, even for increasing vaccination intentions. Our study highlights the importance of testing interventions outside of survey experiments that are lim-

**Data Availability Statement:** The replication package containing the raw data and Stata code can now be accessed via GitHub (https://github.com/IvoSteimanis/covid19_PLOS) and Zenodo (https://zenodo.org/record/6242620). The links

were added to the corresponding fields in the submission mask.

**Funding:** Data collection was supported by a grant from The Deutsche Forschungsgemeinschaft (https://www.dfg.de/en/index.jsp). The grant was received by IS (award number: STE 3083/1-1). The funders had no role in study design, data collection and analysis, decision to publish, or preparation of the manuscript.

**Competing interests:** The authors have declared that no competing interests exist.

ited to measuring vaccination intentions—not actions—and immediate changes in attitudes and intentions—not long-term changes.

## Introduction

The COVID-19 pandemic prompted global research leading to the development of several vaccines in record time. At the same time, countries like Germany are struggling to have enough people vaccinated to relieve pressure on their healthcare systems. The prevalence of certain ideologies in Germany—such as anthroposophy [1], homeopathy [2], and far right-wing supporters [3]–likely provided fertile grounds for skepticism towards COVID-19 vaccines. As of December 2021, almost 27% of Germans were still unvaccinated with vaccination rates plateauing since fall [4]. Meanwhile, vaccine protection is waning [5, 6], and new variants are expected to lead to further waves of infections that could bring public life to a standstill again [7]. To decrease the spread of COVID-19 and the likelihood of the virus mutating, the German government launched information campaigns to highlight the benefits of vaccination and to combat the spread of vaccination myths. We mimic these two types of intervention in a real-world experiment to assess whether such information campaigns are effective in increasing vaccination rates.

Vaccination hesitancy—the delay in accepting or refusing vaccines despite their availability [8]–has been extensively researched prior to COVID-19. Decades of vaccination hesitancy research has shown that hesitancy is complex, multifaceted, and context specific (for an overview see [9]). A large share of variation in hesitancy can be explained by differences in (i) sociodemographic characteristics; (ii) cultural, institutional, and political factors; and (iii) psychological factors [10–15]. However, there is not one universal factor that consistently explains hesitancy as determinants vary not only across countries and vaccine types but also over time.

Research on hesitancy towards COVID-19 vaccines shed further light on the connection of the three broader categories researched in vaccination hesitancy in general. First, sociodemographic characteristics such as age, gender, affluency, and education were associated with vaccination hesitancy in many different contexts [16–24]. Second, cultural, political, and institutional differences such as lower trust in authorities and support of populist views were also found to correlate with higher COVID-19 hesitancy [18, 24–27]. Third, psychological factors such as higher risk appraisal, greater psychological distress from the pandemic, and stronger other regarding preferences have been consistently shown to correlate with lower hesitancy [27–31].

While sociodemographic characteristics as well as cultural, institutional, and political factors are important to explain how hesitancy emerges, it is difficult, sometimes even impossible, to change them. Policies targeting people's attitudes, perceptions, and opinions are likely more fruitful in reducing hesitancy quickly. Moreover, doubts regarding the safety, effectiveness, and benefits of the COVID-19 vaccines on the one hand and an underestimation of the risk of infection and severe illness on the other have been shown to be the most prominent reasons for hesitancy across many countries [24, 26, 28, 29, 32, 33]. One approach to mapping these psychological determinants is the 5C model, which elicits people's Confidence in the safety and efficacy of the vaccine, Complacency about the risk of infection, Constraints that prevent one from vaccinating, Calculation of one's own costs and benefits, and the perceived Collective responsibility to vaccinate [34]. Furthermore, analysis suggests that many psychological factors determining hesitancy towards COVID-19 vaccines are mediated via the 5Cs [31].

One threat to people's confidence in the safety and effectiveness of vaccination is fake news and misinformation. While empirical evidence suggests that exposure to vaccination myths increases vaccination hesitancy [35–37], far less is known about how to counteract the resulting hesitancy. Some studies have found that providing accurate information to debunk vaccination myths increase vaccination intentions [16, 21, 35, 38], while others have found no effect [25, 39]. The impact of emphasizing benefits remains similarly contested, with some studies finding evidence that highlighting benefits increases intentions [38, 40, 41] while others do not [39, 42–45]. These contradicting findings might be explained by country-specific differences at the time of data collection, such as the epidemiological situation or information about the vaccines dominating the news at that time. Furthermore, many studies were conducted before vaccines were available in 2020, and with the exception of Dai et al. [42] all studies cited here had to rely on vaccination intentions instead of actual behavior. To assess whether information campaigns reduce hesitancy, studies are needed that examine their impact on vaccination intention and vaccination actions over time. Beyond these interventions, the convenience of vaccination—such as ease of access and availability—is suggested to reduce hesitancy as well [46, 47]. Moreover, vaccination reminders have been shown to increase COVID-19 vaccination rates [42].

This study investigates how debunking vaccination myths and highlighting benefits in combination with sending vaccination reminders can affect vaccination intentions and actions over time. We expect both debunking myths and highlighting benefits to increase participants' intention to get vaccinated and that repeated debunking of vaccination myths as well as repeated highlighting of vaccination benefits will reduce inaction. First, we ran a preregistered survey experiment in Germany in May/June 2021, testing the effects of debunking vaccination myths and highlighting benefits on vaccination intentions. This was followed by a series of emails reinforcing the information treatments in the survey experiment and concluded with a follow-up survey three months later in September 2021 to measure whether participants were vaccinated. We find that one-time exposure to information, irrespective of the content, does not increase vaccination intentions in the survey experiment. However, communicating vaccination benefits over several weeks increased the likelihood of taking action towards vaccination by 27% compared to the control group, while debunking vaccination myths had no significant effect. Our findings highlight the importance of testing interventions outside of survey experiments that are limited to measuring vaccination intentions—not actions—and immediate changes in attitudes and intentions—not long-term changes. Attitudes, in particular, have been shown to take time to change; see, for example, Albarracin & Shavitt [48] and Bohner & Dickel [49]. In addition, our explorative analysis suggests that participants that did not take any action towards being vaccinated are deeply entrenched in their belief that COVID-19 vaccines are unnecessary and harmful. Only about 10% of these participants reported being vaccinated if financially incentivized, and even fewer reported getting vaccinated if sanctioned.

## Methods

### Experimental design

We conducted a pre-registered experiment in Germany between May and September 2021, with 1,324 unvaccinated participants to measure the extent to which information and reminders can reduce vaccination hesitancy. We proxy vaccination hesitancy by the variables vaccination intention and vaccination inaction described below. The study was implemented in three phases (see Fig 1). From May 25 to June 2, we conducted a survey experiment to examine the effectiveness of debunking vaccination myths (T1: Debunking) or highlighting benefits (T2:

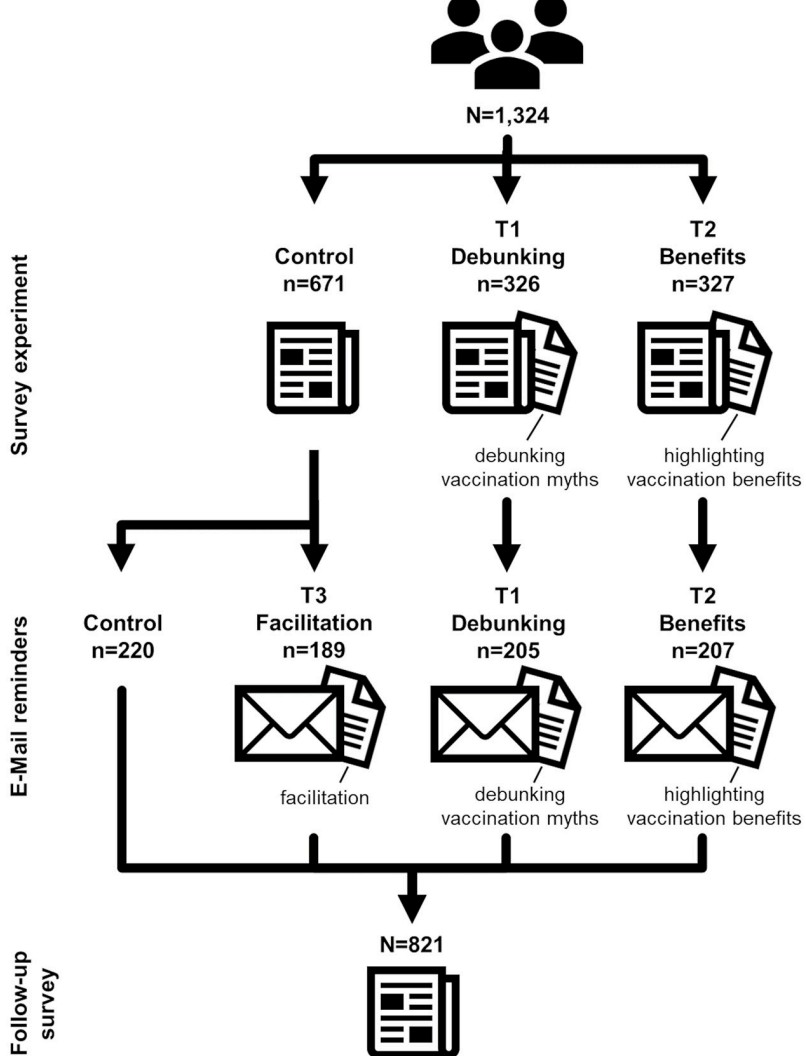

**Fig 1. Study design & treatment groups.** The three study phases are presented in rows, the treatment conditions in columns, and the *n* refers to the number of observations in each phase and treatment group. In the survey experiment, participants were randomly assigned to one of three groups: control, debunking (T1), or benefits (T2). Half of the participants in the control group were then randomly assigned to receiving facilitation emails (T3). Of the 1,324 participants from the survey experiment, 821 (62%) returned the follow-up survey.

Benefits) on vaccination intention. Throughout August, participants in the treatment groups received one email per week, four in total, reminding them to get vaccinated. The emails contained information for participants on where to register for a vaccination appointment and, depending on participants' treatment group, additional information either further debunking vaccination myths or highlighting the benefits of vaccination. In addition, a facilitation treatment (T3: Facilitation) was implemented at this stage in which half of the participants from the control group also received vaccination reminder emails including information on where to register for an appointment. Between September 6 and 18 we conducted a follow-up survey to record participants' vaccination action. We chose this period for our study as restrictions on applications for vaccination appointments were lifted in Germany in May 2021. S3 Appendix

provides more detailed background information on the discussion and implementation of policies in Germany at the time of the study. In the subsequent months, with much of the demand for vaccination met, waiting lists and sometimes even appointments became obsolete in vaccination centers. To ensure that all participants had ample opportunity to vaccinate, we waited until September to complete our experiment through conducting the follow-up survey.

**Treatments.** In the survey experiment (May/June), 1,324 participants were randomly assigned to one of three treatment conditions: Control, T1 Debunking, or T2 Benefits. The exact presentation of information provided in each condition is reported in S12 Appendix. In the debunking treatment, participants were presented with a brief explanation on why COVID-19 vaccines were approved quicker than other vaccines without skipping any steps in the examination phase, what the efficacy of a vaccine means in terms of protection of the vaccinated person, how well the side effects are researched, and why vaccines do not interfere with people's DNA. Participants allocated to the benefits treatment were provided information on the benefits of vaccination including both individual and collective benefits such as being vaccinated protects one from severe COVID-19 infections, helps to protect others who cannot get vaccinated, lifts all contact restrictions and curfews, and eliminates quarantine requirements when traveling. Both information sets were composed to be as similar as possible to each other in appearance and amount of information to be processed. We designed the treatments to mimic campaigns implemented by governments and organizations. The German government—among others—provided corrective information debunking widespread misbeliefs on the COVID-19 vaccination on the official website [50]. In addition, the Federal Ministry for Health of Germany set up a website highlighting the benefits for oneself and others of the vaccination [51]. We cover these two campaigns in the survey experiment and complement them with the facilitation treatment introduced in the follow-up mails. For this, we randomly assigned half of the participants from the control group to the facilitation treatment. The aim of this treatment was to mimic governmental efforts to increase the convenience of vaccination by invitations to (mobile) vaccination centers.

The emails sent between the survey experiment and follow-up survey contained a reminder to get vaccinated, further information regarding the respective treatment of the survey experiment (only T1 Debunking, T2 Benefits), and information on where local vaccination appointments could be made online or by phone. The pure control group did not receive any emails.

**Hypotheses.** The aim of our study was to assess the efficacy of policies aiming to increase vaccination rates both recommended by the literature and implemented by governments at a time when vaccinations became readily available. Based on previous empirical results, we expected both the debunking and benefits treatment to increase participants' intention to be vaccinated in comparison to the control group leading to our hypotheses (survey experiment):

*H1.1*: *Receiving information debunking vaccination myths increases the intention to get vaccinated.*

*H1.2*: *Receiving information highlighting the benefits of the vaccination increases the intention to get vaccinated.*

In addition, we expected that sending vaccination reminders and making people aware of information facilitating vaccination enrollment, would reduce inaction (follow-up survey):

*H2.1*: *Receiving emails providing information where and how to register for vaccination increases the probability of getting vaccinated*

Lastly, we expected that repeated information provision, either debunking vaccination myths or highlighting vaccination benefits, in combination with facilitation will reduce inaction:

*H2.2*: *Receiving debunking information emails increases the probability of getting vaccinated*

*H2.3*: *Receiving emails highlighting the benefits of the vaccination increases the probability of getting vaccinated*

## Measurement variables

Our main outcome variables are (i) intention to get vaccinated and (ii) participants' vaccination action. We measured participants' intention by asking whether they would vaccinate if they had the opportunity on a 7-point Likert scale ranging from "definitely would not" to "definitely would" (see S12 Appendix for experimental materials). In addition, we adopted Betsch et al.'s [34] five psychological antecedents of vaccination scale (5C scale): Confidence that the vaccine is safe and effective, Complacency about the risk of infection, Constraints that prevent one from vaccinating, Calculation of one's own costs and benefits, and the perceived Collective responsibility to vaccinate. In May/June 2021, most vaccination centers administered only vector vaccines (e.g., AstraZeneca and Johnson & Johnson), and some people were waiting for mRNA vaccines (e.g., BioNTech/Pfizer and Moderna) to become available before registering for a vaccination appointment. Therefore, we recorded participants' Confidence in mRNA and vector vaccines separately. Both intention to get vaccinated and the 5C scale were measured after the treatment to avoid demand effects.

To record participants' vaccination action, we asked them if they were fully vaccinated, partially vaccinated, had a vaccination appointment, were on a waiting list for an appointment, or had taken no actions to vaccinate so far. Vaccination action was assessed prior to the treatment in the survey experiment. As participants who reported already being fully or partially vaccinated were excluded from the May/June 2021 survey experiment, only the following actions were recorded: having an appointment, being on a waiting list, or not yet having taken an action. Vaccination action was also assessed in the follow-up survey. From this vaccination action, we deduce the binary variable *inaction* which takes the value 1 if no action was taken and 0 otherwise.

As explanatory variables, we recorded participants' risk-perception, anticipated regret of (not) getting vaccinated, emotional response to COVID-19, experiences with previous vaccinations, and dogmatism. *Risk perception* was assessed in three steps. First, we asked participants a series of questions about the likelihood of infection, the severity of the disease, and long-term consequences, such as long COVID ($\alpha = .71$). Based on these responses we ranked their perception of risk to themselves and others on a scale from 0 "no risk" to 100 "high risk." Participants were then asked to correct their ratings if they felt misrepresented by it. Finally, the average of the two scores was taken to determine the risk perception score used in the analysis (see S6 Appendix for details).

*Anticipated regret* was measured, following Brewer et al. [52], by asking participants whether they would regret vaccination if they later experienced side effects and whether they would regret not being vaccinated if they later became seriously ill with COVID-19. Responses for both questions were recorded on a 7-point Likert scale. To construct the net-anticipated regret score, the score from the first question was subtracted from the score of the second question.

To record participants' *emotional response to COVID-19*, we asked them how strongly they feel upset, alarmed, nervous, attentive, or anxious when thinking about COVID-19 on a scale from 1 "not at all" to 7 "extremely" for each emotion respectively. In the analysis, we control for the average over all five emotions ($\alpha = .88$).

*Dogmatism*–the tendency to rigidly, uncritically adhere to beliefs—was assessed using a condensed version of Altemeyer's [53] dogmatism scale, with answer options ranging from 1 "do not agree at all" to 7 "fully agree". Responses towards the nine items are compiled into an index where higher values are associated with a more dogmatic mindset ($\alpha$ = .78). All measurements were pre-tested for clarity and validity in an online survey with 575 students at the University of Marburg in May 2021 (see S1 Appendix for details).

## Sample

In total, 1,623 people completed the online survey experiment in May/June and 987 participants completed the follow-up survey in September. Participants were recruited by Respondi via its online opt-in panel on the platform Mingle and directed to our surveys. In accordance with Respondi's guidelines, participants received .75 euros per survey, each of which took on average 14 minutes to complete. Surveys were generated using the software SoSci Survey [54] and made available to participants via www.soscisurvey.de. Following our pre-registered exclusion criteria, we excluded participants who rushed through either of the surveys in less than 5 minutes, showed signs of inattention, or reported not having answered all questions in an attentive manner. We also excluded participants with a discrepancy greater than one year in the reported age or specified different genders between surveys. Thus, the final sample includes 1,324 participants for the survey experiment and 821 participants for the follow-up survey. Our results are robust to the exclusion criteria, although excluded participants do differ from the remaining sample in some respects. For example, excluded participants tend to be male, younger, have lower net anticipated regret, and are less likely to intend to vaccinate (see S4 Appendix for details). In the limitations section and S5 Appendix, we discuss how attrition might affect our results.

Due to the requirement of being unvaccinated to participate, it appears that we have a higher proportion of participants from areas with lower vaccination rates. Fig 2A shows where our participants come from, Fig 2B shows the proportion of unvaccinated individuals across Germany on June 2, 2021. Although the origin of our participants and the vaccination rate do not match perfectly, there is a distinct overlap in the areas where we sampled more participants (dark blue) and where many people are unvaccinated (dark red). In addition, our sample reflects the age distribution of the unvaccinated population in Germany at that time (S8 Appendix).

Compared to the general population, we sampled fewer elderly people (aged 60 and above), since they were prioritized for vaccination in the months before data collection. In terms of gender, our sample is not significantly different from that of the general population. Slightly more than 50% of our participants are female and their age structure roughly follows the general age distribution of Germany under 50 years. In our sample (Table 1), over 40% are married and 29% and 26% graduated from high school and university, respectively.

## Statistical analysis

Our study design, data collection, and analysis were pre-registered on AsPredicted (Ref. 66735). We use ordinary least square regression to estimate treatment effects on vaccination intention and action with heteroscedasticity robust standard errors. Due to the binary outcome variable for vaccination action, we used non-linear probability models to estimate treatment effects as robustness checks—reported in S9 Appendix. For the main results, we control for a set of pre-specified covariates to improve the precision of our treatment estimates. These covariates include the participants' socioeconomic status, such as age, highest completed education, marital status, household income adjusted by household size (see S6 Appendix for

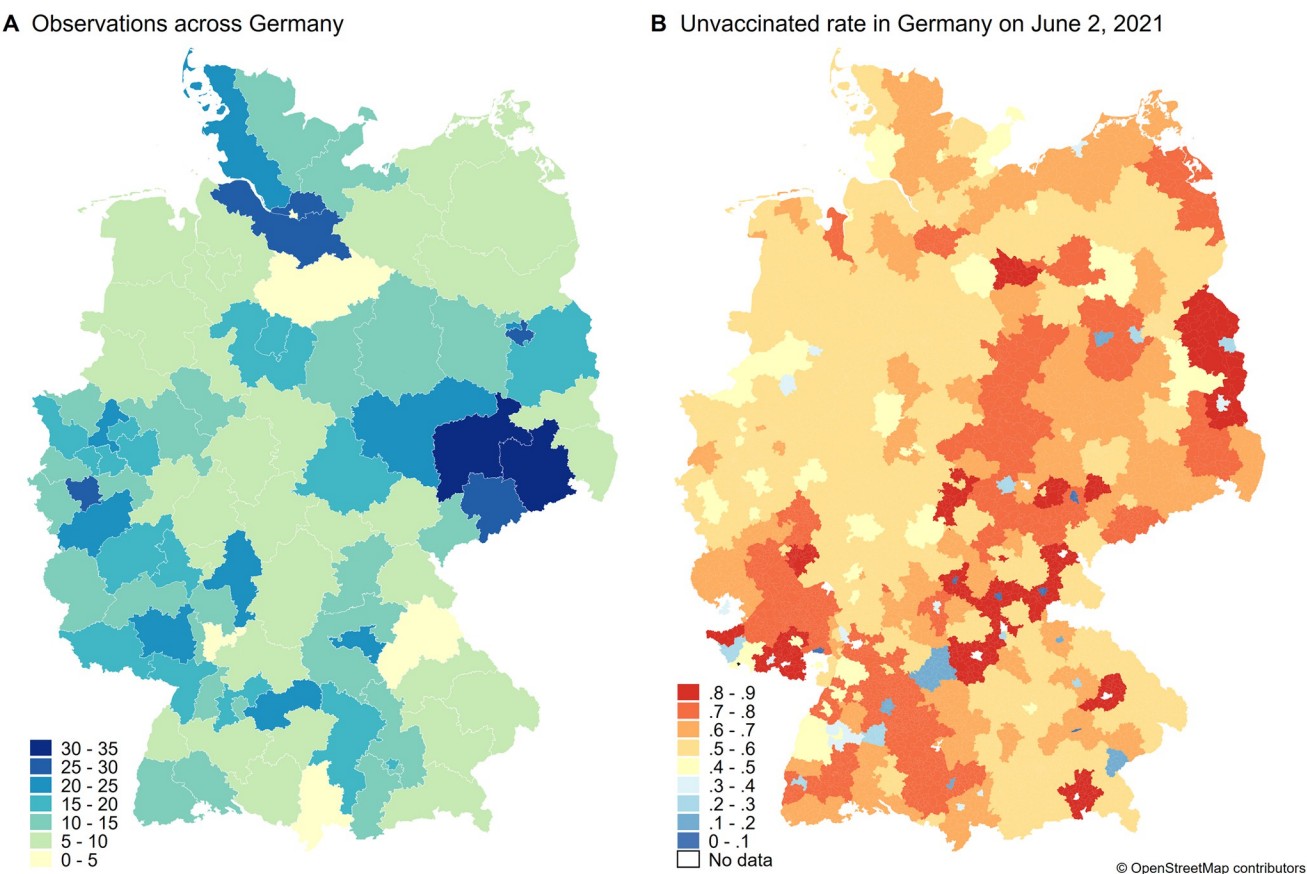

**Fig 2. Sample distribution over Germany.** Panel A shows where study participants are from. We asked participants for the first two digits of their area code and colored them according to the number of participants in each area. Panel B shows the proportion of unvaccinated individuals in Germany as of June 2, 2021. The map was created using publicly available vaccination data from the RKI [55] and shows the proportion of unvaccinated individuals by administrative code (AGS) for which the number of vaccinations was reported. For population data, we use data from the 2011 Census from the Federal Statistical Office Germany [56]. Because area codes and administrative area codes do not completely overlap, there are slight discrepancies in the areas highlighted in panels A and B. Both maps were created with Stata 16 using shapefiles from OpenStreetMap, available under the Open Database (https://www.openstreetmap.org/copyright) [57].

details), perceived health status, and whether participants ever rejected a vaccination before. Based on our power calculation used to determine sample sizes, we should be able to detect small effect sizes for vaccination intention and medium effect sizes for vaccination action. All analyses were conducted with Stata version 16.1.

## Ethics statement

The study received ethical approval from the German Association for Experimental Economic Research e.V. (GfeW, no. IC4Gnr8z). The welcome page of our survey informed participants about the guidelines of ethical empirical research: voluntariness, compensation, benefits, and anonymity. In case they had any questions or concerns they could contact the corresponding author at any time via email (exact wording is reported in S12 Appendix). In addition, participants were provided with clear information regarding the purpose, duration, and scope of the study before giving their written informed consent. Participants had to state that they are (i) 18 or older, (ii) agree to receive up to four emails, (iii) read and understood the information on ethical empirical research, and (iv) are willing to participate in this study. All data were

**Table 1. Summary statistics.**

| | Mean | SD | Min | Max |
|---|---|---|---|---|
| *Outcome variables* | | | | |
| Vaccination intention: mRNA | 5.17 | 2.34 | 1 | 7 |
| Vaccination intention: Vector | 3.32 | 2.33 | 1 | 7 |
| 5C: Confidence mRNA | 4.71 | 1.97 | 1 | 7 |
| 5C: Confidence Vector | 3.91 | 1.86 | 1 | 7 |
| 5C: Constraints | 2.17 | 1.33 | 1 | 7 |
| 5C: Complacency | 3.19 | 1.79 | 1 | 7 |
| 5C: Calculation | 5.08 | 1.60 | 1 | 7 |
| 5C: Collective responsibility | 5.05 | 1.81 | 1 | 7 |
| Vaccination inaction (= 1) | 0.52 | 0.50 | 0 | 1 |
| *Explanatory variables* | | | | |
| Denied other vaccines | 0.14 | 0.35 | 0 | 1 |
| Index: COVID-19 risk perception | 39.41 | 22.46 | 0 | 100 |
| Index: Emotional response to COVID-19 | 3.61 | 1.48 | 1 | 7 |
| Net anticipated regret (no vaccination—vaccination) | 0.95 | 4.00 | -6 | 6 |
| Index: Dogmatism | 3.95 | 0.96 | 1 | 7 |
| *Socio-economic variables* | | | | |
| Female | 0.51 | 0.50 | 0 | 1 |
| Age: <30 | 0.20 | 0.40 | 0 | 1 |
| Age: 30–39 | 0.13 | 0.33 | 0 | 1 |
| Age: 40–49 | 0.22 | 0.41 | 0 | 1 |
| Age: 50–64 | 0.37 | 0.48 | 0 | 1 |
| Age: 65+ | 0.08 | 0.27 | 0 | 1 |
| Secondary school: "Hauptschulabschluss" | 0.11 | 0.32 | 0 | 1 |
| Secondary school: "Realschulabschluss" | 0.35 | 0.48 | 0 | 1 |
| High school: "Fach & allg. Hochschulberechtigung" | 0.28 | 0.45 | 0 | 1 |
| University degree | 0.26 | 0.44 | 0 | 1 |
| HH Income: < 1001€ | 0.11 | 0.32 | 0 | 1 |
| HH Income: 1001€ - 3000€ | 0.50 | 0.50 | 0 | 1 |
| HH Income: 3001€– 4500€ | 0.25 | 0.43 | 0 | 1 |
| HH Income: > 4500€ | 0.13 | 0.34 | 0 | 1 |
| Household members aged 0–14 years | 0.26 | 0.59 | 0 | 4 |
| Household members above 14 years | 1.49 | 1.26 | 0 | 23 |
| Married | 0.41 | 0.49 | 0 | 1 |
| Observations | 1324 | | | |

Notes: Table 1 shows mean, SD, minimum and maximum values of outcome, explanatory, and control variables. More details on sample characteristics are provided in S7 Appendix.

handled confidentially, stored safely, and we ensured that the anonymous survey experimental data are kept separately from non-anonymous payments data.

# Results

The results section is organized as follows. First, we look at treatment effects of one-time messages debunking vaccination myths or highlighting vaccination benefits on vaccination

intentions in the survey experiment (H1.1 and H1.2). Second, we analyze the effects of the repeated information provision by email on vaccination actions reported in the follow-up survey using the balanced panel dataset (H2.1, H2.2, and H2.3).

## Survey experiment: Treatment effects on vaccination intentions and 5C

Our findings show a clear discrepancy in vaccination intentions between mRNA (Pfizer, Moderna) and vector (AstraZeneca, Johnson & Johnson) vaccines. While 50% of participants report that they would definitely be vaccinated with a mRNA vaccine, only 17% would definitely accept a vector vaccine (Fig 3A). On average, 62% of participants intend to get vaccinated (scores above 5) with mRNA vaccines but only 25% do with vector vaccines. For vector vaccines, the share of participants (49%) who do not intend to get vaccinated (score below 3) is more than double compared to mRNA vaccines (21%). Overall, 54% of participants prefer mRNA over vector vaccines, with 44% being indifferent, and 2% stating a preference for vector vaccines. The average intention for vector vaccines is 1.8 points lower on the 7-point Likert scale than for mRNA vaccines (T-Test diff. = 1.85, $t_{1,323}$ = 30.12, p < .001).

In the following analysis of treatment effects, we report only intentions to vaccinate with mRNA vaccines because at the time of our study people could freely choose their vaccines in Germany and almost all participants either had a clear preference for mRNA vaccines or were indifferent. Furthermore, there were no shortages of mRNA vaccines in Germany at that time. Results for intentions to vaccinate with vector vaccines are reported in S9 Appendix.

We start with testing the hypothesis of whether debunking vaccination myths (T1: Debunking) or highlighting the benefits of getting vaccinated (T2: Benefits) increase the intention to get vaccinated using multiple regression (Fig 3B). Contrary to our expectation, we find that neither the debunking treatment (coefficient ß = -.02, p = .819, 95%CI: -.21 to.17) nor the benefits treatment (ß = .11, p = .228, 95%CI: -.07 to.29) significantly increases participants' intentions to get vaccinated. One concern could be that respondents belonging to a priority group are more hesitant than other respondents as they had more time to get vaccinated but are not. Testing for this possibility we find no differences in treatment effects between priority and non-priority groups. Furthermore, respondents belonging to a priority group reported higher intentions than respondents without priority status, not lower. Beyond, we preregistered several heterogeneous effects but do not find any of them to be statistically significant. In all estimations, we additionally control for differences in covariates between treatment groups (see S9 Appendix for additional results and robustness checks).

Further analyses show that socioeconomics are jointly significant but do not explain much of the variation in mRNA vaccination intentions (adj. $R^2$ = .025). Adding individual-level variables, for which we plotted the point estimates in Fig 3B, drastically increases the explained variance in intentions (adj. $R^2$ = .640). Especially net-anticipated regret of getting vaccinated compared to not getting vaccinated is a strong predictor for vaccination intention. A one SD increase in net-anticipated regret increases vaccination intentions by 1.3 points (ß = 1.35, p < .001, 95%CI: 1.24 to 1.45). In addition, participants who have not taken any steps towards getting vaccinated (ß = -1.10, p < .001, 95%CI: -1.29 to -.91) and participants who denied other vaccines in the past (ß = -.52, p < .001, 95%CI: -.78 to -.28) have lower intentions.

The analysis of treatment effects on the 5C is reported in S9 Appendix. The factors captured by the 5C-scale are strongly correlated with the intention to get vaccinated—they jointly explain 74% of the variation in intentions (adj. $R^2$ = .737). Given that the treatments had no effect on the intention it is not surprising that we do not find a consistent effect of the treatments on the 5C.

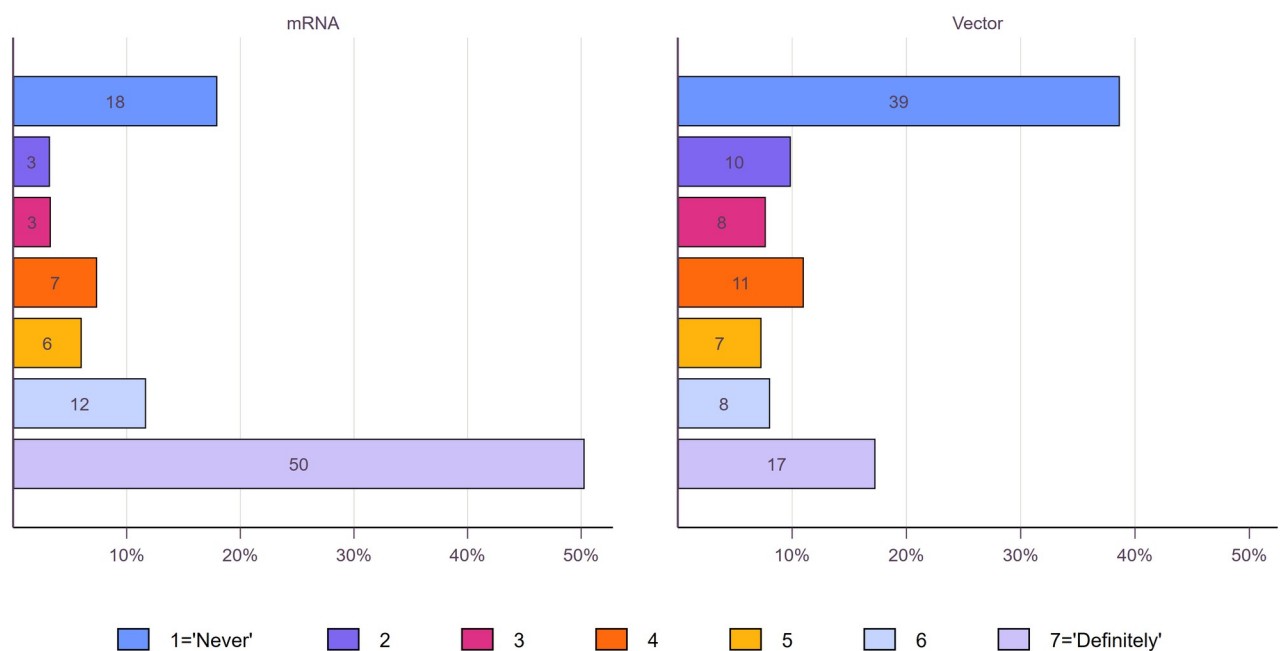

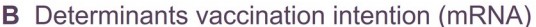

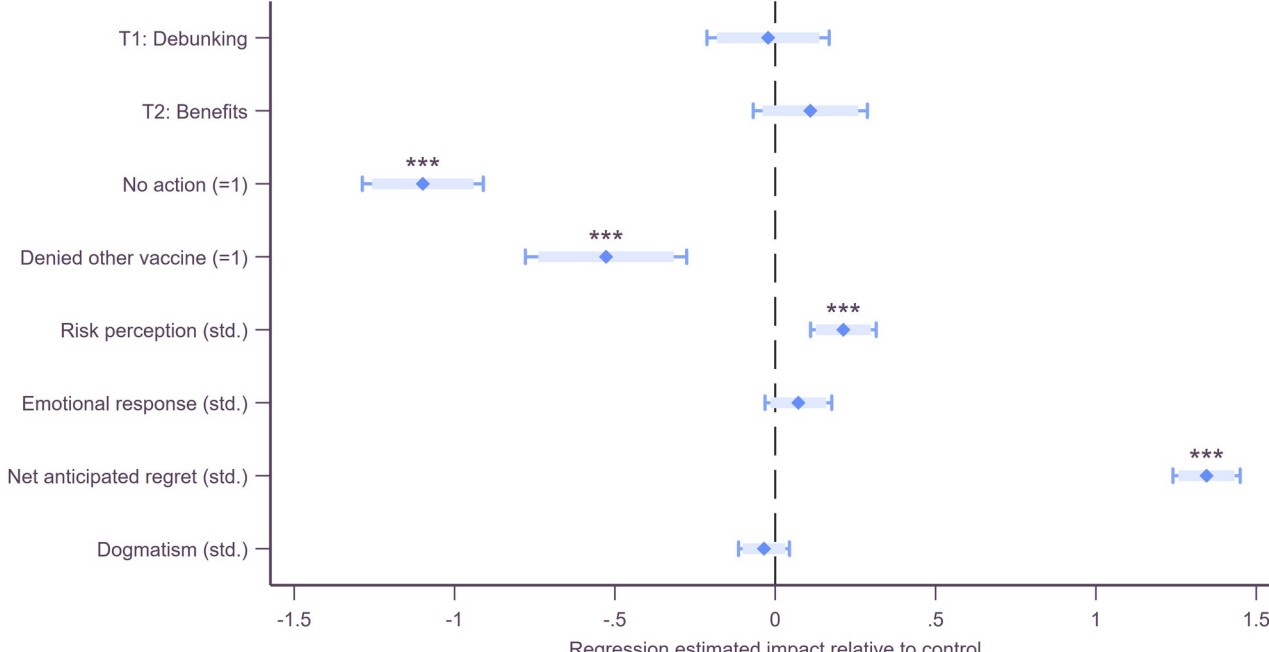

**Fig 3. Vaccination intentions.** Panel A shows vaccination intentions across treatments separately for mRNA and vector vaccines measured using 7-point Likert scales. Panel B plots regression estimates with vaccination intention (mRNA) as the dependent variable from ordinary least square regressions controlling simultaneously for the plotted variables and additionally for gender, age, education, adjusted household income, and marital status. Heteroskedasticity robust standard errors were used to compute 95% (thin bars) and 90% (thick bars) confidence intervals. Full regression outputs are reported in S9 Appendix, model 3 corresponds to the plotted estimates in panel B.

### Follow-up survey: Treatment effects on self-reported vaccination (in)action

Next, we focus our analyses on participants' self-reported actions and test whether our treatments were successful in reducing inaction. For this we use the balanced panel dataset, that is, only include those who have participated both in the survey experiment and follow-up survey. At the time of our first survey, half of the participants (52%) had not taken any actions towards getting vaccinated, 33% were on a waiting list, and only 15% already had an appointment (see Fig 4A). In the follow-up survey, 65% of participants were fully vaccinated, and 28% (n = 233) still had not taken any actions to get vaccinated. Only a few participants either were only partially vaccinated (3%), had an appointment (2%), or were on a waiting list (2%). Given this dichotomous outcome in self-reported vaccination actions in the follow-up survey, we use a binary specification of inaction for the analyses of treatment effects.

Fig 4B shows regression estimates from linear probability models predicting the likelihood of not having taken any action in the second wave. On average (blue estimates), the benefits treatment significantly reduced inaction by about 6 percentage points compared to the control group (ß = -.06, p = .07, 95%CI: -.13 to.00), while neither the debunking of false information (ß = -.01, p = .71, 95%CI: -.08 to.05) nor facilitation alone (ß = -.04, p = .24, 95%CI: -.11 to.03) had any significant effect. However, it seems intuitive that our treatments could only affect those who had not reported any actions in the survey experiment (orange estimates). Almost everyone who had reported to have taken action to get vaccinated in the survey experiment was fully vaccinated in the follow-up survey. Among those participants who had taken no actions to begin with, the benefits treatment reduced inaction by 19 percentage points compared to the control group (interaction ß = -.19, p < .001, 95%CI: -.32 to -.06). Using sample splits, we find that the benefits treatment has equal effects among socio-economic groups. Only women seem to react less to highlighting vaccination benefits than men (see S9 Appendix). Even though the benefits treatment significantly reduced vaccination inaction, there is still a large proportion of participants who had not taken any actions to get vaccinated yet. In the discussion, we explore the determinants of this inaction in greater detail.

### Discussion

Our results suggest that repeatedly highlighting the benefits of getting vaccinated in combination with providing facilitation information decreases vaccination inaction while debunking vaccination myths or providing only facilitation information had no effect. Although many studies have found positive effects of providing factual information about the safety and efficacy of COVID-19 vaccines on reducing vaccination hesitancy, our results, in line with Ashworth et al. [40], do not support this. Given that efficacy and safety concerns regarding the vaccine are the main driver of vaccination hesitancy, our results are surprising at first sight.

There are two plausible reasons why the debunking treatment did not decrease vaccination inaction. First, it could be that the information provided was already known to participants, as vaccine efficacy and safety have been discussed in the public since late 2020. The effect of the debunking treatment on vaccination hesitancy is likely to be much smaller for someone who already feels well-informed than someone to whom this information is new. Indeed, we find that 33% (n = 138) of participants in the debunking treatment reported not being concerned with one of the aspects debunked in the treatment, and 23% (n = 74) were not concerned with any aspect. Second, prior opinions of participants might have been too strong to be changed significantly by the information provided. We find, for example, that participants in the debunking treatment do not feel better informed about the safety and efficacy of vaccines even after reading the information provided (T-Test diff. = .12, $t_{868}$ = .97, p = .333, see S9 Appendix). Furthermore, we asked participants in the debunking treatment to rate on 7-point Likert

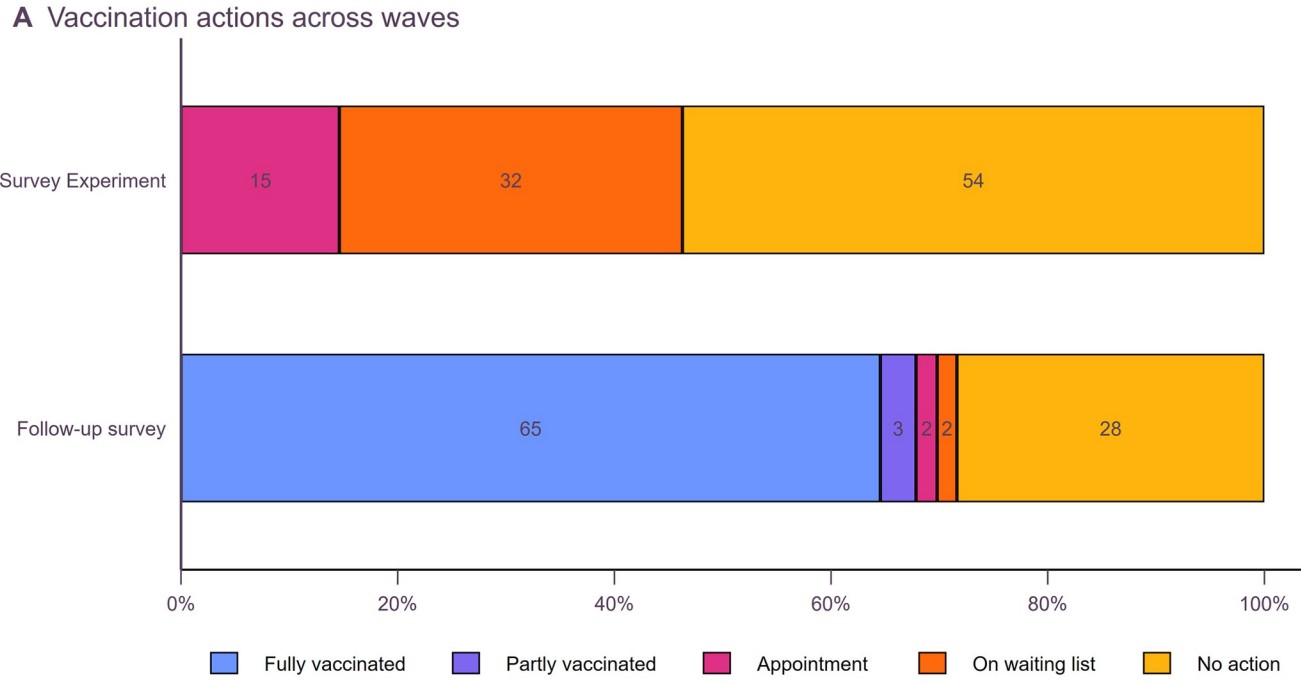

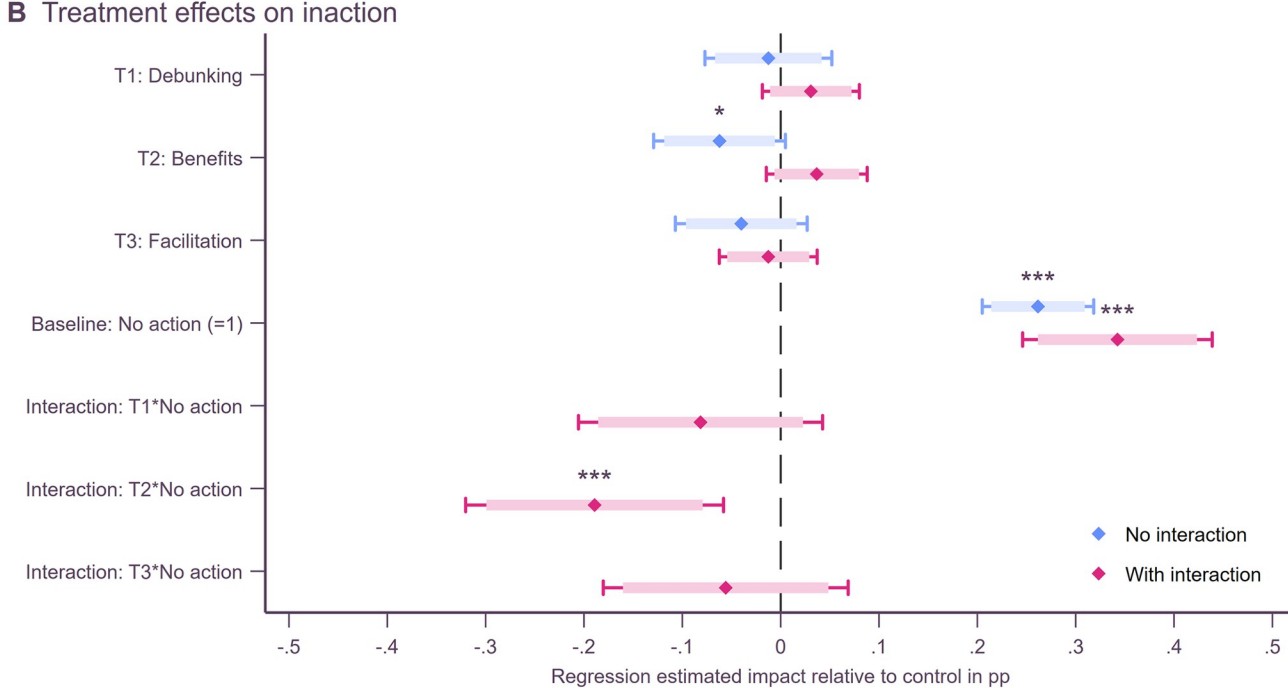

**Fig 4. Self-reported actions of the balanced panel.** Panel A shows the actions taken by participants across both waves. Panel B shows regression estimates with no action in the second wave as the dependent variable from linear probability models controlling for gender, age, education, adjusted household income, marital status, denied other vaccines, COVID-19 risk perception, emotional response, net anticipated regret, and dogmatism. Heteroscedasticity robust standard errors were used to compute 95% (thin bars) and 90% (thick bars) confidence intervals. Full regression outputs and robustness checks using non-linear Probit regression models are reported in S9 Appendix.

items to what degree (1 'not at all' to 7 'completely') the provided information reduced their concerns about the vaccines. Worryingly, concerned participants did not catch up to less concerned participants. They still feel significantly worse informed about the safety and efficacy of vaccines (7-point Likert item) than less concerned participants (T-Test diff. = -1.17, $t_{285}$ = -5.96, p < .001) and have lower vaccination intentions (T-Test diff. = -1.94, $t_{324}$ = -7.92, p < .001). Taken together, about one-third of the participants in the debunking treatment were not concerned regarding the COVID-19 vaccines, and the rest were not strongly convinced by the provided information. That could explain the lack of significant effects on reducing vaccination hesitancy.

In the following, we explore factors that explain why some participants are still not vaccinated and what policies could convince them to get vaccinated. These insights could help to understand the underlying reasons for inaction among these participants, as well as potential interventions to overcome vaccination hesitancy from a policy-makers perspective.

## What determines being unvaccinated?

Participants who reported being on a waiting list or having an appointment in the first survey almost exclusively reported being fully vaccinated in the follow-up survey. Only eight participants who initially reported taking some action did not follow through with their plans to get vaccinated. Thus, in the following, we reduce the sample to only those participants who reported not having taken any actions to get vaccinated in the survey experiment and participated in the follow-up survey (n = 441). Within that sample, we check for correlations of different determinants on the likelihood of still not having any action in the second wave.

Using a stepwise modeling approach shows that socio-economic characteristics are jointly significant predictors of vaccination behavior, but they only explain 3% of the unique variation (adj. $R^2$ = .03). Additionally controlling for the 5Cs increases the explained variation by more than tenfold (adj. $R^2$ = .35). The model controlling for other reasons such as COVID-19 risk perception, emotional response, dogmatism, and anticipated regret in addition socioeconomics explain about 25% of the variation (adj. $R^2$ = .25). However, when all factors are simultaneously controlled for, the model has the same explanatory power (adj. $R^2$ = .35) as the model only controlling for socioeconomics and the 5Cs—indicating the importance of the 5Cs (S9 Appendix). Fig 5 shows the effect each item of the 5Cs measured in the first wave has on self-reported vaccination behavior in the follow-up survey. Among the 5Cs, we find that higher Confidence in mRNA vaccines and Collective responsibility are correlated with a lower probability of inaction. Calculation, the degree to which individuals engage in extensive information seeking to weigh the risks of infection versus vaccination, is correlated with a higher probability of inaction. Participants who perceive COVID-19 as riskier (Complacency) also show slightly lower inaction. However, vaccination inaction does not correlate with structural or psychological barriers (Constraints) in our sample.

While the aforementioned determinants provide useful insights into the underlying factors of inaction, we additionally allowed participants who had not taken any action yet to state their own reasons for not doing so in the follow-up survey (n = 233). Participants could tick pre-defined reasons as well as formulate their own reasons. Fig 6 shows the frequency with which each item was mentioned (size of nodes) and the frequency with which reasons were mentioned in conjunction with each other (size of edges, i.e., links between notes). The resulting network highlights the interconnectivity of reasons given for vaccination inaction. It suggests that participants who believe COVID-19 vaccines are harmful often also indicate that they are unnecessary. In addition, these participants tend to be concerned about the side effects, the amount of research, and share the view that the government and the vaccination

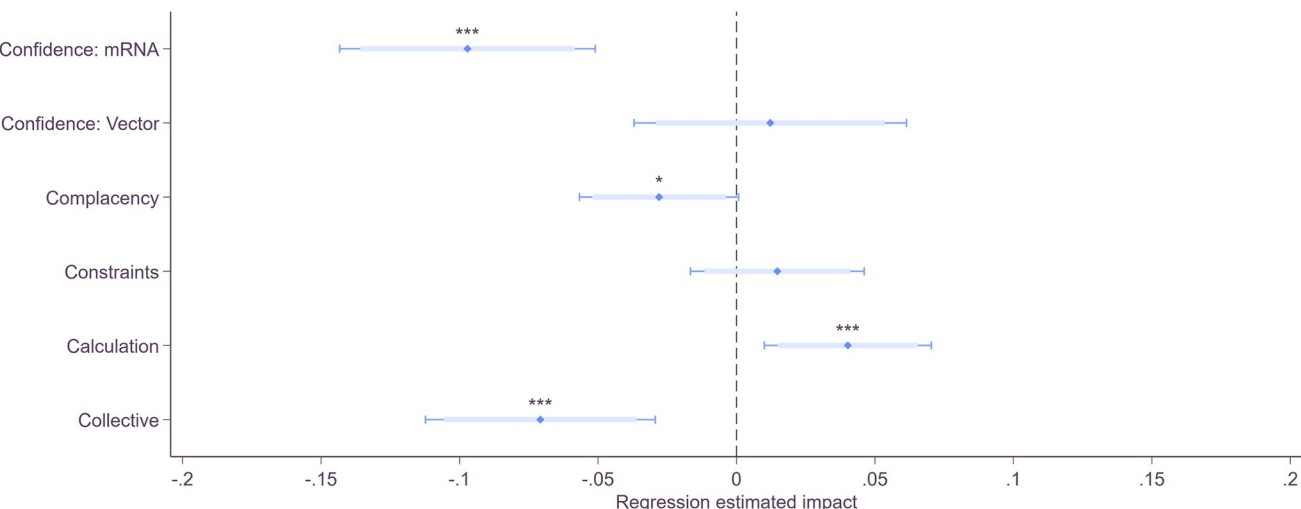

**Fig 5. Vaccination inaction and 5Cs.** Regression estimates for the five psychological antecedents of vaccination with the binary variable inaction in the second wave as the dependent variable from linear probability models are presented. We also control for gender, age, education, adjusted household income, marital status, denied other vaccines, COVID-19 risk perception, negative emotions, net anticipated regret, and dogmatism. Estimates are obtained from multiple least square regressions with robust standard errors: *** p < .01, ** p < .05, * p < .1. Full regression outputs are reported in S9 Appendix.

commission are not trustworthy. Finally, a surprisingly high proportion of participants also said they could not be vaccinated for medical reasons. Whether this is true or a result of over-estimating the side effects, we can only speculate.

Comparing the results from the self-stated reasons (Fig 6) with the correlates of vaccination inaction with the 5C factors (Fig 5) reveals that, if inactive participants are asked directly, they

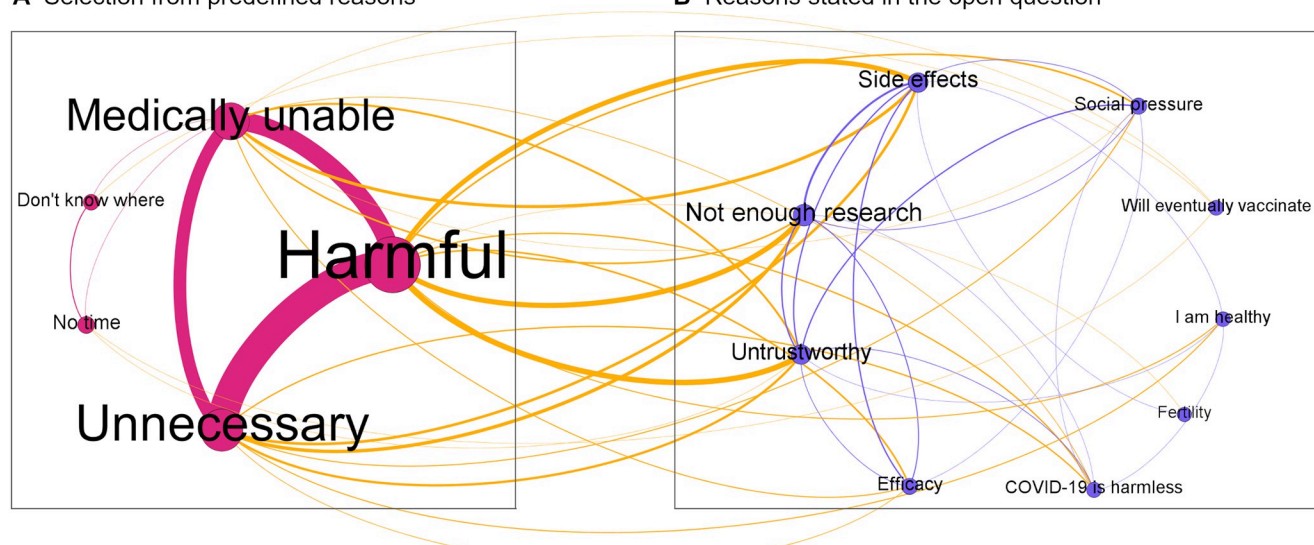

**Fig 6. Reasons stated for vaccination inaction.** Participants were asked about the reasons why they did not vaccinate. We categorized the reasons listed and highlighted which reasons were mentioned together. Panel A shows the relationship between pre-defined reasons participants could choose from in blue. Panel B shows the relationship of reasons that participants additionally mentioned in an open text box in orange. Connections between pre-defined and self-stated reasons are illustrated in green. The network was illustrated with Gephi 0.9.2.

are more likely no name safety concerns (Confidence) and argue that the vaccine is not neces-
sary (Complacency). Yet, the correlates also reveal that inactive participants are more con-
cerned about the costs and benefits (Calculation) of getting vaccinated and less concerned
about the collective well-being (Collective) compared to vaccinated participants. This might
indicate why the benefits treatment but not the debunking treatment reduced inaction. Even if
people could be convinced that COVID-19 vaccines are safe and effective, they nevertheless
might still believe that they are unnecessary. In addition, inactive participants are much more
skeptical towards the government and vaccination commission. Consequently, they are likely
more skeptical towards any information in support of vaccines. Highlighting the benefits of
vaccination might therefore prove to be a more effective approach as it directly affects the cost-
benefit calculation.

Our results suggest that pure information provision—debunking vaccination myths,
highlighting benefits, or facilitation—still leaves room for improvement in terms of reducing
vaccination hesitancy in the population. A significant share of our participants, just as the in
general German population, had not taken any vaccination action towards getting vaccinated
at the time of the second survey. In the following, we explore the potential of further policies
aiming to reduce vaccination hesitancy.

## Incentives to convince the unvaccinated

Prominent policies that have been discussed in politics, tested in research, and/or employed by
some countries are monetary incentivization, facilitation of vaccination (e.g., getting vacci-
nated at home, receiving an invitation for vaccination, getting vaccinated at a shopping cen-
ter), and disadvantages for those unvaccinated (e.g., exclusion from public events, and
discontinuation of free COVID-19 tests). To explore the efficacy of these policies in the Ger-
man context, we asked hesitant participants, who have not taken any actions to get vaccinated
yet, about their (hypothetical) willingness to get vaccinated if certain policies were imple-
mented. Firstly, participants were asked about their willingness to accept a COVID-19 vaccine
if provided a monetary incentive (Fig 7A). We started with an incentive of 50 Euros which
continued to increase every time the participant declined. In line with results on the effective-
ness of monetary incentives for actual vaccination rates from a Swedish sample [58], we find
that providing a modest monetary incentive of 50 Euros increases the acceptance. However,
further increases have relatively little effect compared to the amounts offered, for example, a
tenfold increase in the amount of money offered increased the willingness to vaccinate by a
mere 8 percentage points.

Furthermore, we asked participants whether they would vaccinate if any of the policies
named above were implemented (Fig 7B). While monetary incentives appear to have the great-
est impact, overall, only 13% of hesitant participants could be persuaded to vaccinate with any
incentive listed, which would further increase the vaccination rate in our sample by 3.7 per-
centage points. Although these are only exploratory results, they also indicate resistance to
strong interventions such as exclusion and sanctions. Earlier studies found that strong inter-
ventions find little appreciation, especially among the hesitant [47], and may increase anger
among the hesitant [59] possibly leading to behavior to regain their restricted freedom [45]. It
has therefore been argued that governmental interventions aiming at increasing vaccination
uptake need to be designed with great care [60]. Some participants indeed stated concerns in
the open comment section such as skepticism or resistance towards governmental action: "As
long as the vaccination is advertised with bonus, financial or material, the whole story is sus-
pect to me", "If I have made up my mind about something and did my independent research,
then I don't change my opinion/attitude just because someone sends me something and above

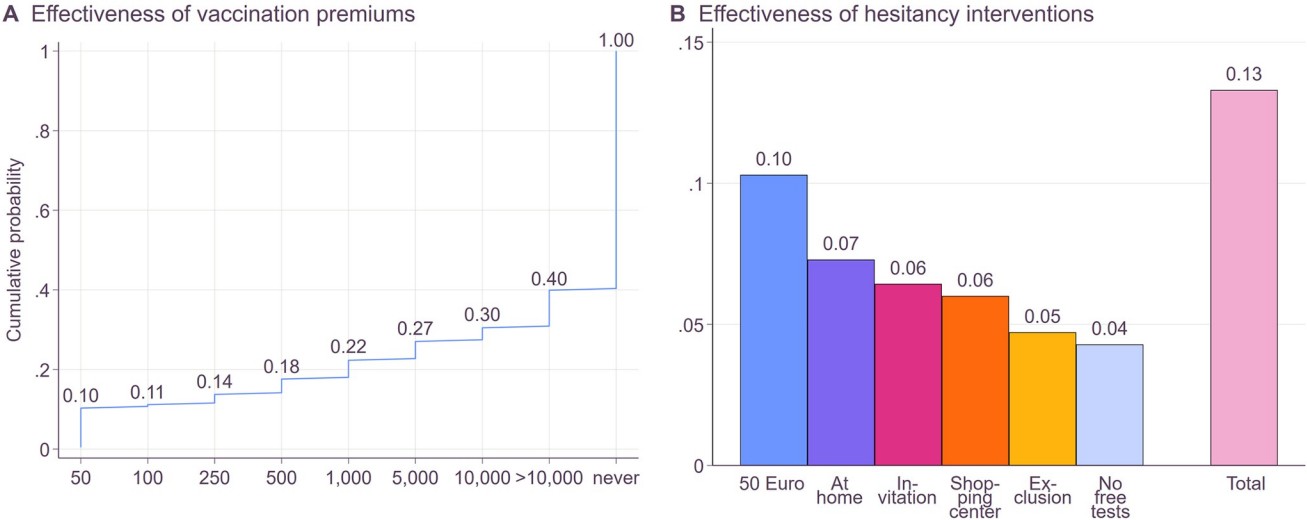

**Fig 7. Effectiveness of interventions to decrease hesitancy.** Panel A shows the cumulative distribution of unvaccinated participants who would vaccinate given a certain vaccination premium. We asked if they would vaccinate if they would receive a 50 Euro premium. If not, we repeated the question and increased the premium sequentially. Panel B shows the share of unvaccinated participants who would vaccinate given a 50 Euro premium if someone would come to their home to vaccinate them if they received an invitation to get vaccinated if they could vaccinate in a shopping center, if unvaccinated people were excluded from public events, or if the government would stop providing free COVID-19 tests. The final column to the right shows the total share of participants who stated an intention to vaccinate in one of the 6 hypothetical interventions.

all thereby wants to talk me into something", "I don't want to be blackmailed" or "The pressure to vaccinate is getting stronger, the advertising for vaccinations is getting weirder, and the supplemental offerings are getting more insane". While others stated in the open comment section that they indeed decided in favor of vaccination due to outside pressure, the possibility that such measures produce resistance should be kept in mind.

## Limitations

The results reported have some limitations. First, the vaccination action measured is self-reported. It has been argued that participants may behave differently once cues about appropriate behavior have been provided by the experimenter, that is, the experimenter demand effect [61]. In our case, participants in the treatment conditions may have sensed that we deemed getting vaccinated as a desirable action and therefore stated to have been vaccinated more often in the treatment groups. Furthermore, if participants understood the aim of the study—that is, testing the efficacy of information provision in reducing vaccination hesitancy —they may have untruly reported being vaccinated [62]. We tried to control for such distortion by asking participants to report the batch number of their vaccines. We find no difference in batch numbers reported across treatments (see S10 Appendix). Furthermore, demand effects are possibly lower in online research where the contact between the experimenter and the participant is minimized. Lastly, the fact that we do not find significant treatment effects on vaccination intention in the survey experiment also hints at the absence of experimenter demand effects.

Second, a substantial part of participants did not return for the second survey (attrition of 38%), potentially threatening the internal validity of our results on vaccination inaction reported in Fig 4. We tested for both differences in attrition rates and selectivity in terms of baseline outcome measures. Regarding attrition rates, we find that participants from the

control, debunk, and benefits groups were equally likely to return, while those in the facilitation treatment were 6 percentage points less likely to return compared to the control group. To further understand whether attrition affects the internal validity of our results, we compare the mean baseline outcomes of control and treatment participants by return status [63], for details see S5 Appendix. These comparisons make us confident that the follow-up results are internally valid as we find that participants who returned have similar baseline outcome values across control and treatment groups.

Third, our results should be carefully situated in the context in which our study was conducted when generalizing to the broader population or even different countries. The treatment interventions started when vaccines became widely available to the entire adult population in Germany (end of May 2021). Thus, our treatments might have produced different results if they were assessed at another time, for example before vaccines were available in 2020. In addition, vaccination rates in Germany (67%) at the time of the follow-up survey were comparable to German-speaking countries (63% and 62% in Austria and Switzerland respectively, EU average: 66%) but much lower compared to other European countries such as Portugal (87%), Spain (80%), or, Denmark (76%) on September 19, 2021 [64]. Our results might therefore, be more transferable to countries with similar contexts, such as Austria and Switzerland. These German-speaking countries have substantial cultural, historical, and economic ties and share a similar system of federalism with mandatory universal health insurance [3, 65, 66]. While all three countries had prompt and similar governmental responses to the COVID-19 outbreak (school closures, obligatory masks, enforce social distancing, free testing facilities, contact tracing, etc.) vaccination rates plateaued by the end of September 2021 despite readily available vaccines [3].

Lastly, the data is from an opt-in online panel provided by Respondi. While it seems representative for the German population in terms of age and gender, this might not be the case for people's willingness to get vaccinated. Indeed, comparing our sample to the rest of Germany reveals that study participants in the balanced panel were on average much more likely to get vaccinated within the study period than the general German population (increase in vaccination rate of 67.84% in our sample vs. 40% in the German population over the same time frame, see S11 Appendix). This could be due in part to selective attrition but we suspect it more likely represents ex-ante differences in the willingness to vaccinate of people in the opt-in Respondi pool compared to the general population.

## Conclusion

Despite these limitations, our results show the potential of repeatedly informing people about the benefits of vaccination and facilitating the process to decrease vaccination hesitancy. Moreover, our explorative results suggest that providing relatively small monetary incentives could help to further increase vaccination rates. These findings could be important for guiding future vaccination campaigns to reduce the spread of and deaths related to COVID-19. Given the large proportion of people not yet vaccinated in many countries, waning vaccine protection, the emergence of new variants, and the requirement of booster vaccinations, the development of effective COVID-19 vaccination campaigns will remain important for the foreseeable future.

## Supporting information

**S1 Appendix. Survey registration, and pre-test.**
(PDF)

**S2 Appendix. Literature review.**
(PDF)

**S3 Appendix. Data collection and study site background.**
(PDF)

**S4 Appendix. Exclusion criteria.**
(PDF)

**S5 Appendix. Attrition.**
(PDF)

**S6 Appendix. Measurement details.**
(PDF)

**S7 Appendix. Balancing and manipulation check.**
(PDF)

**S8 Appendix. Sample details.**
(PDF)

**S9 Appendix. Additional results and robustness checks.**
(PDF)

**S10 Appendix. Demand effects.**
(PDF)

**S11 Appendix. External validity.**
(PDF)

**S12 Appendix. Experimental materials.**
(PDF)

**S1 Fig.**
(TIF)

## Acknowledgments

We thank Moritz Fritschle for his support in the design and operationalization of the study, Maryia Makhnach for her research assistance, our colleagues for their valuable comments on the earlier drafts of the paper, and the participants of the study for their time and effort.

## Author Contributions

**Conceptualization:** Maximilian Nicolaus Burger, Ivo Steimanis.

**Data curation:** Maximilian Nicolaus Burger, Matthias Mayer.

**Formal analysis:** Maximilian Nicolaus Burger, Ivo Steimanis.

**Funding acquisition:** Maximilian Nicolaus Burger, Matthias Mayer, Ivo Steimanis.

**Investigation:** Maximilian Nicolaus Burger, Matthias Mayer.

**Methodology:** Maximilian Nicolaus Burger, Matthias Mayer, Ivo Steimanis.

**Project administration:** Maximilian Nicolaus Burger, Matthias Mayer.

**Software:** Maximilian Nicolaus Burger, Matthias Mayer, Ivo Steimanis.

**Supervision:** Maximilian Nicolaus Burger, Matthias Mayer.

**Validation:** Maximilian Nicolaus Burger, Matthias Mayer, Ivo Steimanis.

**Visualization:** Matthias Mayer, Ivo Steimanis.

**Writing – original draft:** Maximilian Nicolaus Burger, Matthias Mayer, Ivo Steimanis.

**Writing – review & editing:** Maximilian Nicolaus Burger, Matthias Mayer, Ivo Steimanis.

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
