## [Decision Letter · Decision Letter 0]

31 Jan 2022

PONE-D-21-40313Repeated information of benefits reduce COVID-19 vaccination hesitancy: Experimental evidence from GermanyPLOS ONE

Dear Dr. Burger,

Thank you for submitting your manuscript to PLOS ONE. After careful consideration, we feel that it has merit but does not fully meet PLOS ONE’s publication criteria as it currently stands. Therefore, we invite you to submit a revised version of the manuscript that addresses the points raised during the review process. When revising the paper, please consider the reviewers' comments listed at the end of this email.

We look forward to receiving your revised manuscript.

Kind regards,

Camelia Delcea

Academic Editor

PLOS ONE

Journal Requirements:

3. We note that Figure 2 in your submission contain [map/satellite] images which may be copyrighted. All PLOS content is published under the Creative Commons Attribution License (CC BY 4.0), which means that the manuscript, images, and Supporting Information files will be freely available online, and any third party is permitted to access, download, copy, distribute, and use these materials in any way, even commercially, with proper attribution. For these reasons, we cannot publish previously copyrighted maps or satellite images created using proprietary data, such as Google software (Google Maps, Street View, and Earth). For more information, see our copyright guidelines: http://journals.plos.org/plosone/s/licenses-and-copyright.

a) You may seek permission from the original copyright holder of Figure 2 to publish the content specifically under the CC BY 4.0 license.  

Reviewers' comments:

Reviewer's Responses to Questions

**Comments to the Author**

1. Is the manuscript technically sound, and do the data support the conclusions?

Reviewer #1: Yes

Reviewer #2: Yes

2. Has the statistical analysis been performed appropriately and rigorously? 

Reviewer #1: Yes

Reviewer #2: Yes

3. Have the authors made all data underlying the findings in their manuscript fully available?

Reviewer #1: Yes

Reviewer #2: No

4. Is the manuscript presented in an intelligible fashion and written in standard English?

Reviewer #1: Yes

Reviewer #2: Yes

5. Review Comments to the Author

Reviewer #1: The paper is an experimental assessment using some 1,300 participants in Germany in May/June 2021, whether debunking vaccination myths, highlighting the benefits of being vaccinated, or merely sending vaccination reminders decreases hesitancy

This is simply a relevant and competent paper which gives important insights into the effectiveness of different interventions on vaccine hesitancy. The experiment is clearly explained, multiple robustness checks and checks for attrition and demand effects are conducted and presented as supplementary material. The study was pre-registered. In short: the authors know what they write about and investigate effectively and competently.

I only want to point at some minor issues:

1. Figure 3A: do you present percentages here, please make clear.

2. Is Fig3b a multiple regression, were independent variables all included at the same time?

3. Structure is a little unusual, presenting the hypotheses in the methods section. While I understand that the experiment is more clearly explained at this stage, I would advise to restructure and express hypotheses in a separate hypotheses/literature section before the methods.

4. Maybe use capital C’s when explaining 5C model for clarity.

5. On what scale was dogmatism assessed?

6. Briefly explain the platforms and software you use in one sentence, what is Respondi ? And what is ScoSci survey, is it a software?

7. Regarding your outcome measures, sometimes you talk about action, sometimes vaccination hesitancy and sometimes vaccination status. Choosing one specific term would make the text easier to read. Moreover, in measurement section, could you add the timing during which the dependent variables (intention and action) were assessed? Moreover, did you also assess vaccination intention before the treatment or only at the end of the survey? In other words, do you know whether there were differences in vaccination intention before the treatment?

8. In your sample, the older age-groups did already have the chance to get vaccinated as they were part of higher priority groups. This means that these older aged groups only consist of hesitant individuals, while the lower aged groups consist of a combination of non-hesitant and hesitant individuals. Could you conduct additional robustness checks limiting the sample to those that did not get the chance yet to get vaccinated or including interactions with age/priority groups and treatment?

9. There is much recent vaccine hesitancy material out there, also in PlosOne, also using 5C. For instance Wismans et al. (2021), Psychological characteristics and the mediating role of the 5C model in explaining students' COVID-19 vaccination intention, PlosOne, Aug 11;16(8):e0255382.

10. Avoid words like ‘now’. Imagine what that would mean if the paper is read in three year time.

Reviewer #2: Perhaps the chief concern I have with the paper – given its pre-registration, careful design and statistical analysis – and nuanced and transparent analysis of data – is the motivation articulated for the research and the specific interventions.

I know your interventions are also pre-registered and I will not suggest you change those or the hypotheses posed – they are what they are. But I do think you need to do a better job in outlining “state of the art” regarding (a) what we know about vaccine hesitancy in general, (b) what we know about vaccine hesitancy in relations to Covid-vaccination, and (c) also expand on what is known in the political-cultural-institutional setting of this research (Germany, or perhaps German-speaking countries in general, which you argue your results to generalize to). In the below I briefly expand on each of these pointers and why I see revisions needed:

1. Regarding (a) what we know about vaccine hesitancy in general. You cite two of the classic recent overview articles by Dubé and Schmid but you do not provide any generic overview of what is known in general population samples to explain large sources of variation in vaccine uptake or self-reported measures of vaccine hesitancy. I think this should be the basic starting point for any research on the topic. Since there are meta-analyses and literature reviews readily available, condensing that information into a new appendix (summarized shortly in the introduction) or a few sentences of new text in the introduction seems warranted.

2. Regarding what we know about vaccine hesitancy in relations to Covid-vaccination, your current outline merely focus on sociodemographics such as age, gender, and education (Edwards et al., 2021; Neumann-Böhme et al., 2020; Petersen et al., 2021; Ruiz & Bell, 2021). There are of course 100s if not more studies already available, and I certainly agree you cannot summarize all of these, but I am still somewhat confused about the focus solely on these sociodemographics, and why? (are they more generalizeable across countries? Do they explain more variation in general population samples, etc). A further motivation why solely summarizing current knowledge as consisting of these -and not other – factors is needed. I also think you should do a further cursory review looking specifically at large-N (many thousands) studies of general population samples that you judge are high quality studies, and considering including a few other central factors if you deem them relevant. I have no personal pet peeve here, but I am concerned about carefully summarizing “known knowns” and motivating variable foci.

3. Regarding what is known about vaccine hesitancy in relations in the political-cultural-institutional setting of this research (Germany, or German-speaking countries in general), some discussion is needed that (i) vaccine hesitancy is parly demographic (education, age, gender, etc (Schwarzinger et al., 2021; Troiano & Nardi, 2021)) partly psychological (perceptions, fear, personality types, etc (Gerretsen et al., 2021; Murphy et al., 2021; Wismans et al., 2021)) and also (interacting with both demographics and psychology) political-cultural-institutional in its determinants. For example, we know that adherents to specific religious beliefs, though traditions (in your context e.g. The Rudolf-Steiner tradition of Anthroposophy (Fournet et al., 2018)) and political (e.g. right-wing voters in the US and perhaps also in Europe (Ruiz & Bell, 2021)) ideologies express significant differences in vaccination intention and I would guess, also in vaccination uptake. I thus think you need to somehow mention that each country-culture may face rather unique challenges in vaccine uptake, for political-cultural-institutional reasons (in many countries also due to lack of medical infrastructure) and thus contextualize your research better. When you have done so, zooming in on what may be specific in the German-speaking context (there are general population surveys available such as (Graeber et al., 2021) would be proper contextualization of the current study, its design and interventions, and also in discussing the results.

I have carefully read the methods and analyses and have little to complain on. There is a large attrition cross survey waves but it seems to be random, and you handle it well. The statistical analyses are careful and transparent, and also nicely summarized graphically. I especially like the post-hoc tests and the nice graphical summarize of self-reported reasons in Figure 6. Also figure 7 – in summarizing potentially more useful interventions – is very illuminating. Well done!

One minor reservetation though is your dismissal of the rather established 5Cs model. You write that “Fig. 5 shows the effect each item of the 5Cs measured in the first wave has on self-reported vaccination behavior. Interestingly, the model controlling for other reasons such covid risk perception, emotional response, dogmatism, and anticipated regret has the same explanatory power (adjusted R2=0.35) indicating that these factors are already captured by the 5Cs”. I do not agree. Different covariates may explain equal / similar share of variance, even if they do not overlap much in the covariance matrix (the hold out predictors will just be in the residual). To firmly draw the conclusions that your variables covid risk perception, emotional response, dogmatism, and anticipated regret are already captured by the 5Cs variables, I would suggest a mediation analysis. Obviously they are correlated, but I do not understand how you draw the conclusion they are “already captured” by the 5C variables?

I hope you find my comments useful. Good luck in your hard worked research on this important topic!

REFERENCES

Fournet, N., Mollema, L., Ruijs, W., Harmsen, I., Keck, F., Durand, J., Cunha, M., Wamsiedel, M., Reis, R., & French, J. (2018). Under-vaccinated groups in Europe and their beliefs, attitudes and reasons for non-vaccination; two systematic reviews. BMC public health, 18(1), 1-17.

Gerretsen, P., Kim, J., Caravaggio, F., Quilty, L., Sanches, M., Wells, S., Brown, E. E., Agic, B., Pollock, B. G., & Graff-Guerrero, A. (2021). Individual determinants of COVID-19 vaccine hesitancy. PloS one, 16(11), e0258462.

Graeber, D., Schmidt-Petri, C., & Schröder, C. (2021). Attitudes on voluntary and mandatory vaccination against COVID-19: Evidence from Germany. PloS one, 16(5), e0248372.

Murphy, J., Vallières, F., Bentall, R. P., Shevlin, M., McBride, O., Hartman, T. K., McKay, R., Bennett, K., Mason, L., & Gibson-Miller, J. (2021). Psychological characteristics associated with COVID-19 vaccine hesitancy and resistance in Ireland and the United Kingdom. Nature communications, 12(1), 1-15.

Ruiz, J. B., & Bell, R. A. (2021). Predictors of intention to vaccinate against COVID-19: Results of a nationwide survey. Vaccine, 39(7), 1080-1086.

Schwarzinger, M., Watson, V., Arwidson, P., Alla, F., & Luchini, S. (2021). COVID-19 vaccine hesitancy in a representative working-age population in France: a survey experiment based on vaccine characteristics. The Lancet Public Health, 6(4), e210-e221.

Troiano, G., & Nardi, A. (2021). Vaccine hesitancy in the era of COVID-19. Public Health.

Wismans, A., Thurik, R., Baptista, R., Dejardin, M., Janssen, F., & Franken, I. (2021). Psychological characteristics and the mediating role of the 5C Model in explaining students’ COVID-19 vaccination intention. PloS one, 16(8), e0255382.

6. PLOS authors have the option to publish the peer review history of their article (what does this mean?). If published, this will include your full peer review and any attached files.

Reviewer #1: No

Reviewer #2: No

---

## [Author Response · Author response to Decision Letter 0]

10 Mar 2022

Thank you, Camelia Delcea, Reviewer 1, and Reviewer 2, for your comments provided. Some of the points raised by the reviewers echoed issues we had discussed at length prior to the initial submission of the manuscript. In these cases, it was good to get another opinion on matters where we were undecided before. In other cases, the issues raised by the reviewers made us aware of weaknesses in our manuscript we have not noticed before. In both cases, we found the feedback to be consistently constructive, well-founded, and helpful. We are very grateful for the comments provided that helped improve our manuscript.

In the attached file Response to Reviewers, we provide detailed responses to all comments raised by the editor and both reviewers.

---

## [Decision Letter · Decision Letter 1]

21 Apr 2022

PONE-D-21-40313R1Repeated information of benefits reduces COVID-19 vaccination hesitancy: Experimental evidence from GermanyPLOS ONE

Dear Dr. Burger,

Thank you for submitting your manuscript to PLOS ONE. After careful consideration, we feel that it has merit but does not fully meet PLOS ONE’s publication criteria as it currently stands. Therefore, we invite you to submit a revised version of the manuscript that addresses the points raised during the review process.

Specifically, please address reviewer 1's remaining concern.

We look forward to receiving your revised manuscript.

Kind regards,

Jianhong Zhou

Staff Editor

PLOS ONE

Journal Requirements:

Reviewers' comments:

Reviewer's Responses to Questions

**Comments to the Author**

1. If the authors have adequately addressed your comments raised in a previous round of review and you feel that this manuscript is now acceptable for publication, you may indicate that here to bypass the “Comments to the Author” section, enter your conflict of interest statement in the “Confidential to Editor” section, and submit your "Accept" recommendation.

Reviewer #1: All comments have been addressed

Reviewer #2: All comments have been addressed

2. Is the manuscript technically sound, and do the data support the conclusions?

Reviewer #1: Yes

Reviewer #2: Yes

3. Has the statistical analysis been performed appropriately and rigorously? 

Reviewer #1: Yes

Reviewer #2: Yes

4. Have the authors made all data underlying the findings in their manuscript fully available?

Reviewer #1: Yes

Reviewer #2: Yes

5. Is the manuscript presented in an intelligible fashion and written in standard English?

Reviewer #1: Yes

Reviewer #2: Yes

6. Review Comments to the Author

Reviewer #1: Dear authors,

you dealt with all the comments.

The only minor thing missing may be a robustness check without the priority groups. you could with its results in a footnote.

good luck.

you did a fine job

Reviewer #2: (No Response)

7. PLOS authors have the option to publish the peer review history of their article (what does this mean?). If published, this will include your full peer review and any attached files.

Reviewer #1: No

Reviewer #2: No

---

## [Author Response · Author response to Decision Letter 1]

26 Apr 2022

Reviewer #1

Dear authors,

you dealt with all the comments.

The only minor thing missing may be a robustness check without the priority groups. you could with its results in a footnote.

good luck.

you did a fine job

Response: Dear reviewer, thank you for your kind comments and helpful recommendations. We adapted the manuscript in the suggested manner. However, since PLOS ONE does not allow for footnotes, we included the suggested text in the main manuscript reading:

“One concern could be that respondents belonging to a priority group are more hesitant than other respondents as they had more time to get vaccinated but are not. Testing for this possibility we find no differences in treatment effects between priority and non-priority groups. Furthermore, respondents belonging to a priority group reported higher intentions than respondents without priority status, not lower. Beyond, we preregistered several heterogeneous effects but do not find any of them to be statistically significant (see S9 Appendix). In all estimations, we additionally control for differences in covariates between treatment groups (see S9 Appendix for additional results and robustness checks).”

Furthermore, Table S11 displaying the regression of the subsample of respondents without priorization status was added to Appendix S9.

---

## [Decision Letter · Decision Letter 2]

15 Jun 2022

Repeated information of benefits reduces COVID-19 vaccination hesitancy: Experimental evidence from Germany

PONE-D-21-40313R2

Dear Dr. Burger,

We’re pleased to inform you that your manuscript has been judged scientifically suitable for publication and will be formally accepted for publication once it meets all outstanding technical requirements.

Kind regards,

Mohamed F. Jalloh, PhD, MPH

Academic Editor

PLOS ONE

Additional Editor Comments (optional):

Reviewers' comments:

Reviewer's Responses to Questions

**Comments to the Author**

1. If the authors have adequately addressed your comments raised in a previous round of review and you feel that this manuscript is now acceptable for publication, you may indicate that here to bypass the “Comments to the Author” section, enter your conflict of interest statement in the “Confidential to Editor” section, and submit your "Accept" recommendation.

Reviewer #1: All comments have been addressed

2. Is the manuscript technically sound, and do the data support the conclusions?

Reviewer #1: Yes

3. Has the statistical analysis been performed appropriately and rigorously? 

Reviewer #1: Yes

4. Have the authors made all data underlying the findings in their manuscript fully available?

Reviewer #1: Yes

5. Is the manuscript presented in an intelligible fashion and written in standard English?

Reviewer #1: Yes

6. Review Comments to the Author

Reviewer #1: You did a great job, congratulations.

7. PLOS authors have the option to publish the peer review history of their article (what does this mean?). If published, this will include your full peer review and any attached files.

Reviewer #1: No

---

## [Editor Report · Acceptance letter]

20 Jun 2022

PONE-D-21-40313R2 

Repeated information of benefits reduces COVID-19 vaccination hesitancy: Experimental evidence from Germany 

Dear Dr. Burger:

I'm pleased to inform you that your manuscript has been deemed suitable for publication in PLOS ONE. Congratulations! Your manuscript is now with our production department. 

Kind regards, 

on behalf of

Dr. Mohamed F. Jalloh 

Academic Editor

PLOS ONE